# A Neural Network Approach to Identify Glioblastoma Progression Phenotype from Multimodal MRI

**DOI:** 10.3390/cancers13092006

**Published:** 2021-04-21

**Authors:** Jiun-Lin Yan, Cheng-Hong Toh, Li Ko, Kuo-Chen Wei, Pin-Yuan Chen

**Affiliations:** 1Department of Neurosurgery, Keelung Chang Gung Memorial Hospital, Keelung 204, Taiwan; lisa2017@cgmh.org.tw; 2Department of Chinese Medicine, College of Medicine, Chang Gung University, Taoyuan 333, Taiwan; kuochenwei@cgmh.org.tw; 3Department of Radiology, Chang Gung Memorial Hospital at Linkou, Taoyuan 333, Taiwan; eldomtoh@cgmh.org.tw; 4Department of Neurosurgery, Chang Gung Memorial Hospital at Linkou, Taoyuan 333, Taiwan

**Keywords:** glioblastoma, MRI, tumor progression, radiomics, machine learning, neural network

## Abstract

**Simple Summary:**

Glioblastoma is the most common malignant primary brain tumor and has a poor prognosis with inevitable recurrence or progression. The phenotypes of its progression patterns can be diverse, which may potentially affect the treatment plan and clinical outcome. Our study aimed to identify its progression pattern before surgery by using multimodal MRI. The results showed the different progression phenotypes are clinically important, and by using quantitative MR radiomics, together with neural network-based imaging analysis, we can predict glioblastoma progression phenotypes preoperatively.

**Abstract:**

The phenotypes of glioblastoma (GBM) progression after treatment are heterogeneous in both imaging and clinical prognosis. This study aims to apply radiomics and neural network analysis to preoperative multimodal MRI data to characterize tumor progression phenotypes. We retrospectively reviewed 41 patients with newly diagnosed cerebral GBM from 2009–2016 who comprised the machine learning training group, and prospectively included 18 patients from 2017–2018 for data validation. Preoperative MRI examinations included structural MRI, diffusion tensor imaging, and perfusion MRI. Tumor progression patterns were categorized as diffuse or localized. A supervised machine learning model and neural network-based models (VGG16 and ResNet50) were used to establish the prediction model of the pattern of progression. The diffuse progression pattern showed a significantly worse prognosis regarding overall survival (*p* = 0.032). A total of 153 of the 841 radiomic features were used to classify progression patterns using different machine learning models with an overall accuracy of 81% (range: 77.5–82.5%, AUC = 0.83–0.89). Further application of the pretrained ResNet50 and VGG 16 neural network models demonstrated an overall accuracy of 93.1 and 96.1%. The progression patterns of GBM are an important prognostic factor and can potentially be predicted by combining multimodal MR radiomics with machine learning.

## 1. Introduction

Glioblastoma (GBM) is the most common primary malignant brain tumor in adults [1]. Despite multimodal treatment combining surgery, chemotherapy, and radiation therapy, GBM will most often progress within 7–9 months after treatment, and its tumor progression patterns can be diverse [2]. These various types of progression can have different clinical outcomes; for example, a multifocal progression or ventricular spread showed a worse overall survival [3]. However, conventional magnetic resonance imaging (MRI) can only provide limited information that can be used to distinguish different progression patterns before treatment.

Standard MRI sequences for the evaluation of GBM include T2-weighted, fluid-attenuated inversion recovery (FLAIR), pre-gadolinium T1-weighted, and post-gadolinium T1-weighted [4]. However, a study showed that in areas of normal conventional T1 and T2 MRI signals, there were still cancer cells (false negative rate = 16% and 4%, respectively) [5]. Moreover, semantic features that have limited measurable aspects, such as lesion volume and signal intensity, can only provide limited information for analysis. On the other hand, radiomics can be used to convert traditional imaging information into high-dimensional quantitative “big data” [6], which can be applied in most imaging modalities, including computed tomography (CT), and MRI. Radiomic analysis can generate more agnostic features, including first-order, second-order, or other higher dimensional features. First-order features can be represented by a histogram analysis of the voxels in the regions of interest (ROIs). Second-order texture analysis, firstly introduced by Haralick in 1973 [7], can provide information about image homogeneity, contrast, linear structure, and complexity. Higher order analysis includes filter grid or wavelet analysis of the images. The application of filters (such as Laplacian of Gaussian bandpass) can extract specific imaging structures depending on the width of the filter. Other higher order texture analyses, such as model-based and transform-based methods, were also used to maximize the imaging features [8].

These quantitative imaging features can be used as imaging biomarkers to understand tumor progression and clinical outcomes. Using random survival forest analysis, MR radiomics has shown a fair prediction accuracy [9]. More sophisticated methods, such as cancer imaging phenomics, also show promising results for the predicting of progression-free survival (PFS) and distal recurrence patterns [10]. In addition, combining radiomics with neural network-based analysis is an emerging technique in imaging analysis and has been shown to predict the location of GBM progression/recurrence from preoperative multimodal MRI data with an accuracy of nearly 80% [11].

Therefore, a better understanding of the GBM progression pattern and its impact on the clinical presentation is needed. This study aims to analyze preoperative multimodal MRI radiomic data and establish a prediction model for GBM progression patterns by using machine learning and neural network-based imaging analysis.

## 2. Materials and Methods

### 2.1. Patient Inclusion and Exclusion Criteria

This was a combined retrospective and prospective study. Patients included in this study were organized into two groups. First, we included 41 patients retrospectively from 2010 to 2016 as an initial training group for the prediction model of the tumor progression pattern. Another 18 patients were recruited prospectively from 2017 to 2019 for the validation of the model. All patients were identified and discussed at the multidisciplinary neuro-oncology team meeting. Informed written consent was obtained from all patients. This project was approved by the Chang Gung Medical Foundation Institutional Review Board (IRB approval number: 201601862B0) and was conducted according to the World Medical Association Declaration of Helsinki.

The inclusion criteria included adult patients with a newly diagnosed supratentorial GBM suitable for maximal safe resection and temozolomide chemoradiotherapy. All patients underwent complete preoperative and immediate postoperative MRI (within 72 h) evaluations and had at least two follow-up sessions in which definite tumor progression was noted. Pseudo progression was carefully reviewed and was not included in our cohort. The exclusion criteria were having a history of a previous major operation of the brain and being unable to undergo MRI examination for reasons such as claustrophobia or having non-MRI compatible implants.

### 2.2. MRI Acquisition and Imaging Processing

All patients received a standard diagnostic 1.5 or 3T MRI evaluation, including T1-weighted imaging with contrast enhancement, T1-weighted imaging without contrast, T2-weighted imaging, FLAIR, diffusion tensor imaging (DTI), and diffusion-weighted imaging (DWI) within one week before surgical intervention. All other subsequent MRI evaluations included nonvolumetric T1-weighted imaging with contrast enhancement, T2-weighted imaging, FLAIR, and DWI. After surgical resection, an immediate postsurgical MRI was performed within 72 h. Approximately one month after the operation, in patients suitable for concurrent chemoradiotherapy (CCRT), an additional MRI was performed for radiotherapy planning. Follow-up MRI studies were then planned every three months after treatment, with some modification according to the patients’ condition or clinical needs.

Preprocessing of the MRI data includes brain extraction, DTI analysis, imaging coregistration, and ROI definition. The detailed MR imaging process is shown in Appendix A. In short, DTI images were processed using the FSL [12] function to generate output, including fractional anisotropy (FA), tensor eigenvalues, mean diffusivity (MD), DTI isotropic (DTI-p) maps, and anisotropic DTI (DTI-q). Apparent diffusion coefficients (ADCs) were generated directly from the scanner. All MRI data were coregistered to preoperative T1-contrast enhanced MRI data [13,14].

ROIs were defined by the contrast-enhanced lesions in the preoperative T1 contrast-enhanced MRI data. These manually selected ROIs were performed using 3D slicer (version 4.8.2, http://www.slicer.org; accessed on 20 December 2017) [15]. The tumor progression patterns, defined by the T1W contrast enhancement, were categorized according to the following different definitions:Diffuse versus localized patterns: the diffuse progress pattern is defined as continuous progression ≥ 2 cm from the primary resection margin;Distal progression is defined as a separate progression 2 cm beyond the margin;Ventricular spread showed the progression of the tumor with contrast enhancement along the ventricular wall.

### 2.3. Radiomic Analysis and Machine Learning

The 3D volumetric radiomic features were calculated using 3D slicer. The preoperative T1W contrast-enhanced lesion was manually drawn as input volume to calculate the radiomic features. Ninety-two radiomic features, including first-order, second-order, and filtered wavelet analysis, were extracted from 9 different MR sequences. An additional 13 shape features were also used for analysis. Therefore, the total number of features was 841. Detailed features are listed in Appendix A. Feature selections were chosen by using a t-test for the progression patterns. To establish the prediction model, supervised machine learning included supporting vector machine (SVM), K-nearest neighbor (KNN), and decision tree models were used to evaluate the training group. The machine learning was done by using MATLAB (The MathWorks, Inc., Natick, MA, USA, Version r2019a, classification learner/Statistics, and machine learning toolbox). The input variables were 841 features from 41 cases, and the output variables were binary which indicated the diffuse/localized progression pattern. In addition, 20% of the training group was set as cross-validation. The resultant model from the training group was further applied to the prospective external validation group to test the prediction accuracy of the model.

### 2.4. Neural Network Approach for Imaging Phenotype Prediction

A neural network approach was used to predict imaging phenotypes of GBM progression. A total of 2313 MR images were resliced from preoperative T1W contrast-enhanced MRI scans into 448 × 448 pixels in the format of a portable network graphic (png) file. These 2313 images were divided randomly into a training group (1850 slices, 80% of the total data) and a validation group (463 slices, 20% of the total data). The two pretrained models ResNet50 and VGG16 (pretrained on ImageNet, http://www.image-net.org/, accessed on 10 October 2019) were used as base models. Additional layers, including the global average pooling layer, dense layer (512 neurons, rectified linear unit (ReLU) activation), and output layer (2 neurons, sigmoid activation), were added. Results were further visualized by using gradient class activation mapping (Grad-CAM) with the Keras visualization toolkit (https://raghakot.github.io/keras-vis/, accessed on 24 March 2021).

## 3. Results

### 3.1. Patient Characteristics

Forty-one patients were included as the initial training group, and another 18 patients were prospectively included as the external validation group. The general characteristics are shown in Table 1.

Among the 41 training group patients, 33 were male and 8 were female. The mean age at initial diagnosis was 57.4 ± 13.4 years old. The preoperative tumor size, which was defined by the contrast-enhanced lesion, was 42.7 ± 24.4 mL. Gross total resection (GTR) was achieved in 26 out of 41 patients. The median PFS and overall survival (OS) were 182 and 463 days, respectively. Comparable patient characteristics were observed in the validation group (Table 1). Note that of the 18 individuals in the validation group, only 8 died during the study follow-up.

### 3.2. The Clinical Impact of the GBM Progression Pattern

The progression patterns were classified as described in 2.2 MRI acquisition and imaging processing. Twenty-eight out of 41 patients had diffuse progression patterns, while 13 of them progressed locally. There were 5 distal progression patients; 12 exhibited ventricular progression, and 29 showed progression in more than 2 vectors from the resection margin.

The clinical impact of the progression patterns was analyzed in the 41 patients in the training group and is shown in Table 2. The patients with diffuse progression had a longer PFS period (median = 189.5 days versus 136 days, *p* = 0.02) but a shorter OS period (median = 363 days versus 668 days, *p* = 0.032). Patients with ventricular progression had a shorter OS period (median = 354 days versus 180 days, *p* = 0.05). Distal progression showed a better OS period (median = 558 days versus 449.5 days, *p* = 0.01).

### 3.3. Radiomic Analysis and the Prediction Model of GBM Progression

In our study, the diffuse/local progression pattern had the greatest impact on the clinical outcome. Therefore, 153 of 843 features were selected prior to machine learning by using a *t*-test for the prediction of the diffuse progression pattern. All radiomic features were standardized. These features were used for the supervised machine learning model, which included logistic regression, SVM, KNN, and decision tree components. The results are listed in Table 3. The overall accuracy of predicting diffuse/local progression pattern was 81% (range: 77.5–82.5%, AUC = 0.83–0.89) in different models. In our results, the logistic regression and the KNN models showed the best prediction accuracy.

For further evaluation, different trained models were used to assess the 18 prospectively included patients as external validation. The best overall accuracy was found by using the ensemble tree model with an overall accuracy of 72.2%, positive predict value = 68.8%, and negative predict value = 100%. The results of the external validation are shown in Table 4.

### 3.4. The Neural Network Approach for the Identification of the MR Progression Phenotype

A total of 2313 T1 postcontrast MR images containing lesions were included for training. The modified pretrained ResNet-50 model showed an overall accuracy of 93.1%. The sensitivity was 94.4%, and the specificity was 89.9% (Figure 1A). The modified pretrained VGG16 model for classification of the progression pattern had 95.8% overall accuracy, 96.9% sensitivity, and 94.2% specificity (Figure 1B). The VGG16 model classification results are shown in Figure 2.

Figure 2 shows examples of the result obtained by using gradient class activation mapping to demonstrate the visualization of the neural networks. The localized pattern is shown on the left side, and the diffuse pattern is shown on the right side.

## 4. Discussion

In this study, we established a prediction model for the classification of the tumor progression phenotype from preoperative MRI data using a quantitative radiomics machine learning model and convolutional neural network imaging analysis; moreover, we showed the capability of this technique to identify tumors that may progress diffusely.

The progression pattern of GBM has been studied widely. In 1983, Burger et al. [16] focused on the imaging and pathology presentation of GBM and found that most of the tumor progression occurred in the surgical resection margin. An earlier study of 12 cases also showed that 1/3 of patients had distal progression during the follow-up period [17]. To date, similar results can be seen in other studies: 88% of GBMs can progress near the primary location, known as local progression, and approximately 12% of the tumors can have distal progression [18]. Doner et al. [19], in 2013, classified the tumor progression pattern in GBM patients who received surgical resection plus Gliadel wafer into local, diffuse, multifocal, and distal types. In our study, 28 (68.3%) out of 41 patients in the training group and 11 (61.1%) out of 18 individuals in the validation group had a diffuse progression pattern during the follow-up period.

Different patterns of GBM progression have also been shown to be associated with clinical outcomes. In our results, both PFS and OS were significantly different between the diffuse and local progression patterns. PFS was better in the diffuse progression group, and OS was worse in the diffuse progression group. These findings were compatible with previous studies [20]. In Sheriff’s study, the authors found that the time to progression was shorter with a local progression pattern than with a contralateral distal progression pattern (median = 8 months versus 15 months) [21]. Bonis et al. [18] showed that PFS was shorter in patients with local progression than in those with distal progression (9 months versus 21 months, *p* = 0.05). In addition, our results showed a better prognosis in the distal progression group, which is similar to previous studies [20]. Many studies focus on the tumor subventricular zone, demonstrating a worse prognosis in those with ventricular involvement [22]. However, we found patients with ventricular involvement had a longer PFS. However, the mechanism for an extended PFS period remains unclear and is beyond the scope of this study.

Since the diffuse progression pattern showed the most clinical impact, we aimed to predict this progression model using radiomics and tested it using various machine learning models. Our results showed a fair prediction accuracy of 82.5% in the training group and a 72.2% accuracy (tree model) in the validation group. A recent study by Kazenrooni et al. [10] using the Cancer and Phenomics Toolkit (CaPTk) with the SVM model also showed modest prediction accuracy on distal recurrence patterns (AUC 0.56–0.88). Two of the main issues in machine learning are acquiring an adequate sample size for training and having enough qualitative input features. Our radiomics results provided a reasonable sample size of the features for the training model. Although our sample size was limited, a total of 842 features coded in every ROI still provided a reasonable training size that could be used to train the model.

In contrast to radiomic analysis, the convolutional neural network is an emerging technique for imaging analysis. By using the pretrained neural network, our study achieved 93.1–96.6% accuracy in the classification of the tumor progression phenotype. Furthermore, by using the visualization Grad-CAM, we were able to show the visualization result of the neural network; however, further study is needed to obtain more substantiated deep learning results. The neural network has been studied as a surrogate for various biomarkers. Chang et al. [23] classified IDH-1 mutation, 1p/19q codeletion, and MGMT promoter methylation status in malignant glioma using a convolutional neural network with overall accuracies of 94%, 92%, and 83%, respectively. Li et al. [24] achieved an accuracy of 94.4% when identifying an IDH-1 mutation status in low-grade glioma. Other studies have also utilized deep learning algorithms for tumor response assessment [25,26].

There are several limitations in this study. First, the sample size (both training and validation group) was relatively small for the classic machine learning, which may decrease the reliability of the machine learning result. Therefore, we used 2D sliced images in our neural network model for a larger sample size (*n* = 2313). Second, we were not able to consider the timing of the MRI scan in this study; therefore, there was the potential for a temporal sampling error. GBM progression may change over time, and we defined the progression pattern by using the first true progression image compared to the reference image. This approach may have underestimated the progression in the local progress group in our training cohort. Lastly, only 14 (23.7%) patients had the methylation status of the MGMT promoter.

## 5. Conclusions

In conclusion, the progression patterns of GBM are an important prognostic factor, especially the diffuse progression phenotype. Radiomics analysis from multimodal MRI can provide a substantial amount of quantitative imaging biomarkers for imaging analysis. Further application of the machine learning model and deep learning neural network can potentially predict the tumor progression pattern from preoperative MRI data.

## Figures and Tables

**Figure 1 cancers-13-02006-f001:**
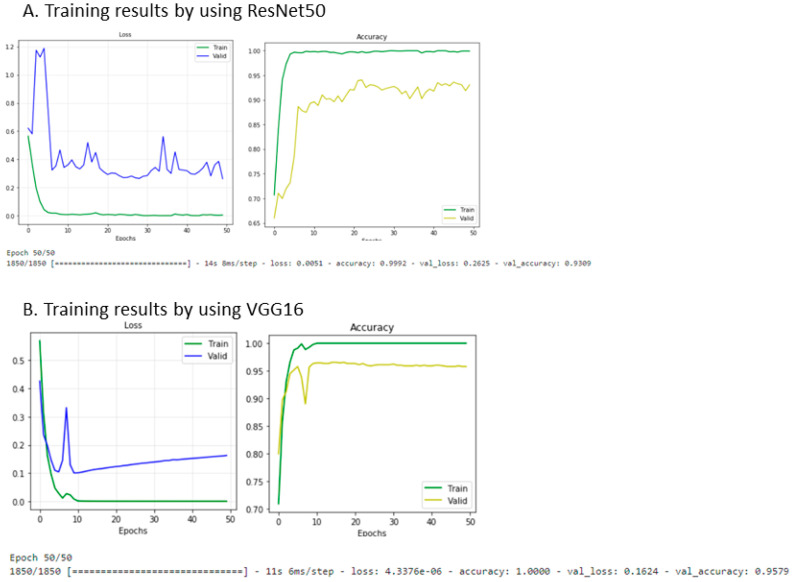
Results of the neural network learning for classification of the progression pattern. This figure shows the results of neural network learning to classify the “diffuse” progression pattern. The training results of the modified pretrained ResNet50 are shown in (**A**), with the changes in loss (left) and accuracy (right) indicated after every epoch. The results of the modified pretrained VGG16 model are shown in (**B**).

**Figure 2 cancers-13-02006-f002:**
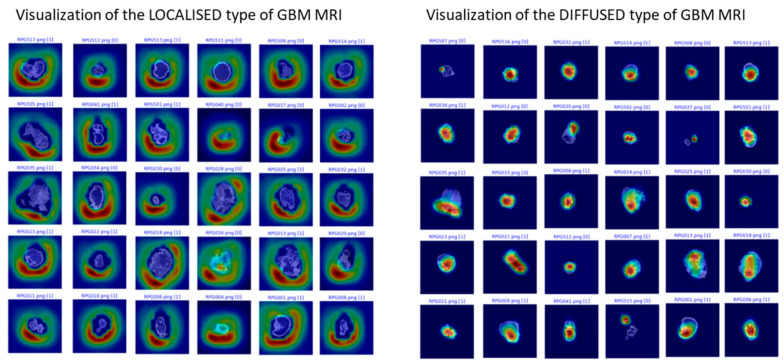
Visualization of the neural network classification result.

**Table 1 cancers-13-02006-t001:** General characteristics.

Characteristics	Training	Validation Group	*p*-Value
Total number of patients	41	18	-
Males/females	33/8	12/6	0.32
Age (years)	57.4 ± 13.4	56.8 ± 11.9	0.87
Pre-OP tumor size (mL)	42.7 ± 24.4	36.19 ± 19.46	0.42
* GTR/STR	26/15	11/7	1.00
PFS (median, days)	182	169.5	0.11
OS (median, days)	463	362.5 * (8 died)	0.68
MGMT unmethylated	1	8	-
methylated	3	2	-
IDH-1 wild type	20	17	0.25
mutated	3	0	-

GTR, gross total resection; STR, subtotal resection; PFS, progression-free survival; OS, overall survival; MGMT, O^6^ -methylguanine-DNA methyltransferase; IDH-1, isocitrate dehydrogenase 1. * The OS were calculated from 10 of the 18 patients in validation group.

**Table 2 cancers-13-02006-t002:** The clinical impact of the progression patterns.

Progression Pattern	Number	Overall Survival (Median, Days)		Progression Free Survival (Median, Days)	
Diffuse	39	363	*p* = 0.032	189.5	*p* = 0.02
Local	20	668	-	136	-
Ventricular spread	22	354	*p* = 0.05	190	*p* = 0.12
No ventricular spread	37	180	-	182	-
Uni-direction	20	490	*p* = 0.66	185	*p* = 0.98
Multidirection	39	449.5	-	173	*-*
Distal	10	558	*p* = 0.01	185	*p* = 0.19
No distal progression	49	449.5	-	173	-

**Table 3 cancers-13-02006-t003:** The outcome of the MR radiomics prediction model in training group.

Train Model	Overall Accuracy	Sensitivity	Specificity	AUC
Linear SVM	77.5%	84.6%	64.3%	0.89
Regression	82.5%	85.7%	75%	0.84
KNN	82.5%	85.7%	75%	0.88
Boosted trees	80.0%	82.8%	72.2%	0.83

SVM, supporting vector machine; KNN, K-nearest neighbor; AUC, area under the curve.

**Table 4 cancers-13-02006-t004:** Outcome of the MR radiomics prediction model in the external validation group (*n* = 18).

Machine Learning Models	Results																		Accuracy
True	1	0	1	1	1	1	0	0	1	1	0	0	1	1	0	0	1	1	Ground Truth
Logistic regression	1	1	1	1	1	0	1	1	1	1	1	0	0	1	1	1	1	1	55.6%
SVM	1	0	1	1	1	1	1	1	1	1	1	0	1	1	1	1	0	0	61.1%
Tree	1	0	1	1	1	1	1	1	1	1	1	0	1	1	1	1	1	1	72.2%
KNN	1	1	1	1	1	1	1	1	1	1	1	1	1	1	1	1	1	1	61.1%

True refers to the true pattern of progression during follow-up; 1: “diffuse” progression pattern; 0: “localized” progression pattern; SVM, supporting vector machine; KNN, K-nearest neighbor.

## Data Availability

Available upon reasonable request from corresponding authors.

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
