# Peer review of "A Neural Network Approach to Identify Glioblastoma Progression Phenotype from Multimodal MRI"

_cancers, 2021, doi:10.3390/cancers13092006_

Round 1

Reviewer 1 Report

This is a nice study that explores progression of glioblastoma post surgery and chemoradiotherapy, classifying it into diffuse and localized. This pattern can be predicted preoperatively by the authors methods, and predicts progression free and overall survival.

It would be interesting to know whether the models could predict ventricular spread, although I note that this feature did not quite reach significance.

It would also be interesting to explore the impact of IDH-1 and MGMT status on progression pattern, but I suspect that the numbers are too small.

There are some English language and style suggestions and areas requiring clarification included in the attached highlighted pdf.

Author Response

Dear reviewer,

Thank you for the review and all the comments. We have revised according to the comments and we would like to respond in a point-by-point manner as follows.

Q1. It would be interesting to know whether the models could predict ventricular spread, although I note that this feature did not quite reach significance.

Reply

In our study, we found the clinical impact of ventricular spreading is less important than diffuse progression pattern. Although the overall survival of patients with ventricular spreading is worse (p = 0.05), the progression free survival did not reach the statistical significance. Therefore, we focused on the prediction of diffuse pattern. However, we did able to predict the ventricular spread phenotype by using “Bagged Tree” and “Boosted Tree” machine learning model from the radiomics features, the accuracy were about 73.7-78.9% which were not shown in the article.

Q2. It would also be interesting to explore the impact of IDH-1 and MGMT status on progression pattern, but I suspect that the numbers are too small.

Reply

It is true that the MGMT methylation status may had impact on the progression patterns1. However, one of the limitations in our study is the lack of the full MGMT and IDH-1 status in all patients. We have added this limitation in the latest revised manuscript. Further survey in these 14 patients with MGMT promoter methylation status, we found no correlation with the progression pattern. 3 out of 5 methylated patients, and 7 out of 9 unmethylated patients progressed diffusely (chi square with Fischer exact p =0.5804). In addition, 25 out of 40 IDH-1 wild type, and 3 all IDH-1 mutated patients progressed diffusely (chi square with Fischer exact p =0.5406). 

Q3. There are some English language and style suggestions and areas requiring clarification included in the attached highlighted pdf.

Reply

Thank you for the suggestions, and we have changed the English and style as suggested. Few comments from are answered as below:

  1. How was pseudoprogression delineated from tumour progression? What MR modalities or additional imaging (PET?) were used?

Reply: We reviewed the whole follow up MRI to exclude those with pseudoprogression which may have radiologic remission later. Therefore, a true progression was confirmed according to the RANO criteria and a continuous radiologic progression on its follow up imaging if obtained.

  1. How did you define what is "tumor progression" is this T1-W contrast enhancement?

Replay: yes

Reference:

  1. Brandes AA, Tosoni A, Franceschi E, et al. Recurrence pattern after temozolomide concomitant with and adjuvant to radiotherapy in newly diagnosed patients with glioblastoma: correlation With MGMT promoter methylation status. Journal of clinical oncology : official journal of the American Society of Clinical Oncology 2009; 27: 1275-1279. DOI: 10.1200/JCO.2008.19.4969.

Reviewer 2 Report

Radiomics is known as the quantitative extraction of subvisual data from conventional radiographic imaging; it is now consider a powerful data-driven approach to offer improvements  into neuro-oncology field particularly referred to diagnosis, prediction, prognosis and treatment response. This paper offer a clinical view- radiomics labeled to a diffuse pathology as glioblastoma with some good points to stimulate the community to grow up in foreseeing how these tumors could grow only through quantitative based imaging. Results are encouraging but not totally documented by the results: radiomics features are not described accurately (supplementary 2 is not available at website www.mdpi.com/xxx/s1) and consequently the parameters considered ( why only progression pattern?). Nevertheless, it's unclear which MRI scan among mentioned, is considered for the evaluation for each group. 

I therefore recomend this paper for publication for the innovations cointained and the aim, for the future perspectives it hides and for the implementations can give to the literature.

Author Response

Dear reviewer,

Thank you for the review and all the comments. We have revised according to the comments and we would like to respond in a point-by-point manner as follows.

Q1. Results are encouraging but not totally documented by the results: radiomics features are not described accurately (supplementary 2 is not available at website www.mdpi.com/xxx/s1) and consequently the parameters considered ( why only progression pattern?).

Reply:

We have uploaded all the radiomics features in this study in the Supplementary 2 according to the instruction of the website. We will confirm with the editor for the detailed information to make sure all supplementary documents will be accessed. Our aim of the study was to identify and predict the progression pattern from the pre-operative MR. We firstly try to find out the importance of the progression pattern. In section 3.2, we have identified “Diffuse” progression had the most clinical impact. Secondly, we conducted radiomics analysis using machine learning model for the identify and prediction of such progression phenotype. Lastly, a different neural network approach was incorporated into the study for the improvement of the prediction model.

Q2. Nevertheless, it's unclear which MRI scan among mentioned, is considered for the evaluation for each group. 

Reply:

The preoperative MRI scans for analysis were obtained from the last scan before surgery, which included 9 different MR sequences (T1W, T1W+C, T2, FLAIR, DTI-q, DTI-p, ADC, MD, FA). The progression MRI was defined as the first radiologic progression MRI (according to the T1W+C). Detailed was described in section 2.2.

Reviewer 3 Report

The authors analyzed 41 patients with newly diagnosed cerebral GBM, representing the base for a machine learning training group, then they prospectively included 18 new patients data validation, to  create a machine learning model and neural network-based model to identify GBM progression patterns

Such models could represent valuable add to a recent field of interest about AI and machine learning applied to neuro-oncology, whereas I have different issues.

The first one is related to sample size: are 18 patients enough to validate a such complex model? How was this sample size calculated, to assess its statistical strenght?

Lines 119-123: "the tumor progression patterns were categorized according to the following different definitions:

1. Diffuse versus localized patterns: the diffuse progress pattern is defined as continuous progression ≥ 2 cm from the primary resection margin. 2. Distal progression is defined as a separate progression 2 cm beyond the margin. 3. Ventricular spread showed the progression of the tumor with contrast enhancement along the ventricular wall".

Is this an arbitrarly choosen definition? Or there are some supporting references, maybe based on computational models?

The main issue of such purely "descriptive" models is, in my opinion, that lot of these models does not take into account biochemical and mechanical interactions between to tumor and its local micro-environment. Does the present model could be flexible enough to comprise this information? (see i.e. Colombo MC et al, Towards the Personalized Treatment of Glioblastoma: Integrating Patient-Specific Clinical Data in a Continuous Mechanical Model.  PLoS One. 2015;10:e0132887 OR Acerbi F. et al. (2021) Mechano-Biological Features in a Patient-Specific Computational Model of Glioblastoma. In: Seano G. (eds) Brain Tumors. Neuromethods, vol 158. Springer, New York, NY. https://doi.org/10.1007/978-1-0716-0856-2_12)

Minor issues:

ref 1 (Ostrom et al., CBTRUS statistical report: primary brain and central nervous system tumors diagnosed in the United States in 2007-2011. 325 Neuro Oncol 2014) it should be better to cite the last version of the CBTRUS, updated to 2013-2017 and published in 2020 (Ostrom QT et al, CBTRUS Statistical Report: Primary Brain and Other Central Nervous System Tumors Diagnosed in the United States in 2013-2017. Neuro Oncol. 2020 Oct 30;22(12 Suppl 2):iv1-iv96. doi: 10.1093/neuonc/noaa200)

Author Response

Dear reviewer,

Thank you for the review and all the comments. We have revised according to the comments and we would like to respond in a point-by-point manner as follows.

Q1. The first one is related to sample size: are 18 patients enough to validate a such complex model? How was this sample size calculated, to assess its statistical strenght?

Reply:

There was no optimal ratio between training sample size and validation sample size. However, 70/30 or 80/20 are mostly used[1]. Therefore, 18 patients were prospectively enrolled based on the retrospective collected 41 patients (18 / (41+18) = 30.5%). We did not use the internal validation, which is commonly used, but such small sample size may cause over-fitting.

One of the limitations of this study was the relatively small sample size, which can be the future work to test on a larger sample size. Traditionally, it is best to have sample size larger than 80 (we had only 59)[2]. However, several methods can be used to overcome the small sample problem, such as increase training feature followed by feature selection[3].

Q2. Lines 119-123: "the tumor progression patterns were categorized according to the following different definitions: 1. Diffuse versus localized patterns: the diffuse progress pattern is defined as continuous progression ≥ 2 cm from the primary resection margin. 2. Distal progression is defined as a separate progression 2 cm beyond the margin. 3. Ventricular spread showed the progression of the tumor with contrast enhancement along the ventricular wall". Is this an arbitrarly choosen definition? Or there are some supporting references, maybe based on computational models?

Reply

Patterns of tumor progression has been described in different article and with diverse description[4,5]. The rationale for defining “Diffuse” type as progression ≥ 2 cm was comparative to “localised” which was often defined by progression within 2cm or within the adjuvant radiation therapy clinical target volume[6,7]. The definition of distant progression was still various from study, but according to our previous systemic review, “a separate progression 2 cm beyond the margin” was the most common definition with a definite descriptive distance [5] . The definition of the ventricular spreading was also described in the systemic review.

Q3. The main issue of such purely "descriptive" models is, in my opinion, that lot of these models does not take into account biochemical and mechanical interactions between to tumor and its local micro-environment. Does the present model could be flexible enough to comprise this information? (see i.e. Colombo MC et al, Towards the Personalized Treatment of Glioblastoma: Integrating Patient-Specific Clinical Data in a Continuous Mechanical Model.  PLoS One. 2015;10:e0132887 OR Acerbi F. et al. (2021) Mechano-Biological Features in a Patient-Specific Computational Model of Glioblastoma. In: Seano G. (eds) Brain Tumors. Neuromethods, vol 158. Springer, New York, NY. https://doi.org/10.1007/978-1-0716-0856-2_12)

Reply

It is true that local environment is important to the tumor progression. As these two valuable articles suggested that GBM cell tend to invade alone fibers which can be represented by diffusion tensor imaging. Such mechano-biology study was based on the established mathematical model. In my knowledge, these two models were different from the machine learning or neural network basis. One of the advantages of the deep learning technique is the omit of the “weight” of the features which allow us to include 9 different MR sequences for analysis.

On the other hands, I believe the model can be improved by adding more clinical or biological features in the future.

Minor issues:

Q4. ref 1 (Ostrom et al., CBTRUS statistical report: primary brain and central nervous system tumors diagnosed in the United States in 2007-2011. 325 Neuro Oncol 2014) it should be better to cite the last version of the CBTRUS, updated to 2013-2017 and published in 2020 (Ostrom QT et al, CBTRUS Statistical Report: Primary Brain and Other Central Nervous System Tumors Diagnosed in the United States in 2013-2017. Neuro Oncol. 2020 Oct 30;22(12 Suppl 2):iv1-iv96. doi: 10.1093/neuonc/noaa200)

Reply

Thank you for the suggestion, we had made the correction.

Reference

  1. Nguyen, Q.H.; Ly, H.B.; Ho, L.S.; Al-Ansari, N.; Le, H.V.; Tran, V.Q.; Prakash, I.; Pham, B.T. Influence of Data Splitting on Performance of Machine Learning Models in Prediction of Shear Strength of Soil. Mathematical Problems in Engineering 2021, 2021, 15, doi:10.1155/2021/4832864.
  2. Figueroa, R.L.; Zeng-Treitler, Q.; Kandula, S.; Ngo, L.H. Predicting sample size required for classification performance. BMC Med Inform Decis Mak 2012, 12, 8, doi:10.1186/1472-6947-12-8.
  3. Golland, P.; Grimson, W.E.L.; Shenton, M.E.; Kikinis, R. Small Sample Size Learning for Shape Analysis of Anatomical Structures. In Proceedings of the Medical Image Computing and Computer-Assisted Intervention – MICCAI 2000. MICCAI 2000. Lecture Notes in Computer Science, 2000; pp. 72-82.
  4. Bette, S.; Barz, M.; Huber, T.; Straube, C.; Schmidt-Graf, F.; Combs, S.E.; Delbridge, C.; Gerhardt, J.; Zimmer, C.; Meyer, B.; et al. Retrospective Analysis of Radiological Recurrence Patterns in Glioblastoma, Their Prognostic Value And Association to Postoperative Infarct Volume. Sci Rep 2018, 8, 4561, doi:10.1038/s41598-018-22697-9.
  5. Piper, R.J.; Senthil, K.K.; Yan, J.-L.; Price, S.J. Neuroimaging classification of progression patterns in glioblastoma: a systematic review. Journal of Neuro-Oncology 2018, 139, 77-88, doi:10.1007/s11060-018-2843-3.
  6. Sherriff, J.; Tamangani, J.; Senthil, L.; Cruickshank, G.; Spooner, D.; Jones, B.; Brookes, C.; Sanghera, P. Patterns of relapse in glioblastoma multiforme following concomitant chemoradiotherapy with temozolomide. Br J Radiol 2013, 86, 20120414, doi:10.1259/bjr.20120414.
  7. Brandes, A.A.; Tosoni, A.; Franceschi, E.; Sotti, G.; Frezza, G.; Amista, P.; Morandi, L.; Spagnolli, F.; Ermani, M. Recurrence pattern after temozolomide concomitant with and adjuvant to radiotherapy in newly diagnosed patients with glioblastoma: correlation With MGMT promoter methylation status. J Clin Oncol 2009, 27, 1275-1279, doi:10.1200/JCO.2008.19.4969.

Reviewer 4 Report

The authors used a neural network approach to identify glioblastoma progression phenotype from multimodal MRI.

The following questions should be answered:

  • Explain the GBM with IDH1 mutation. Were these tumors secondary GBMs?
  • Progression occurs after first surgical removal. Did the authors try to identify those tumors which have a high likelihood to recur?
  • Did the authors consider to contrast their prediction with cases showing radiation changes in the setting of differential diagnosis between GBM recurrency and radiation changes?
  • Did the authors consider the midline shift in their progression patterns?

Author Response

Dear reviewer,

Thank you for the review and all the comments. We have revised according to the comments and we would like to respond in a point-by-point manner as follows.

Q1. Explain the GBM with IDH1 mutation. Were these tumors secondary GBMs?

Reply:

Yes for glioblastoma with IDH-1 mutation can be referred as secondary GBM. Two major pathogenesis of GBM has been proposed, primary and secondar. The secondary GBM accounts for 10% of all GBM. It occurs more in young patients (~44 years old) and locates more in frontal lobe. It has a better prognosis, compared to primary GBM. The most common is the point mutation in codon 132 of IDH-1 or less commonly in codon 172 of IDH-2) is the most distinctive markers of secondary GBM(1).

Q2 Progression occurs after first surgical removal. Did the authors try to identify those tumors which have a high likelihood to recur?

Reply:

Almost all GBM progress or recur eventually. In this article all patients had tumor progress/ recur during the study period. Previous studied had found the prognostic factor associated with a longer progression free survival or overall survival, such as extent of resection of the tumor, CCRT, MGMT prompter methylation status, IDH-1 mutation and other biomolecular markers. Since this study focused on the progression pattern, we had shown that the “Diffuse” progression pattern had the worst impact on the overall survival and progression free survival. However, on the other hand, weather such imaging study can be used to predict time to progression can potentially be another interested topic in the future.

Q3 Did the authors consider to contrast their prediction with cases showing radiation changes in the setting of differential diagnosis between GBM recurrency and radiation changes?

Reply

It is true that the radiation necrosis is difficult to differentiate from true tumor progression. Previous study had shown that multimodal MRI together with PET can be used for the differential diagnosis (2, 3). Although radiation necrosis was not the target in this study, the use of deep learning can potentially be a tool for such complex imaging analysis. Gao et al., in 2020 had published their study using light weighted deep neural network and showed the accuracy was 0.903(4). I believed that with the improvement of the algorithm and integration of PET scan information, the accuracy can potentially be better in the future.

Q4. Did the authors consider the midline shift in their progression patterns?

Reply

The patterns of GBM progression can be diverse. According to our previous systemic review on the GBM progression, there were 10 most cited progression phenotypes ((5) Figure2). The use of localised/local, distant/ distal, diffuse, multifocal and subventricular zone were the 5 most cited patterns. Therefore, in this study we focused on the diffuse/ local, subventricular zone, distal and multidirectional pattern. However, it can be interested to use midline shift as one of the progression patterns. Or to test what kind of the tumor had most peritumoral edematous area.  

Reference:

  1. Parsons DW, Jones S, Zhang X, Lin JC, Leary RJ, Angenendt P, et al. An integrated genomic analysis of human glioblastoma multiforme. Science. 2008;321(5897):1807-12.
  2. Abbasi AW, Westerlaan HE, Holtman GA, Aden KM, van Laar PJ, van der Hoorn A. Incidence of Tumour Progression and Pseudoprogression in High-Grade Gliomas: a Systematic Review and Meta-Analysis. Clin Neuroradiol. 2018;28(3):401-11.
  3. Zikou A, Sioka C, Alexiou GA, Fotopoulos A, Voulgaris S, Argyropoulou MI. Radiation Necrosis, Pseudoprogression, Pseudoresponse, and Tumor Recurrence: Imaging Challenges for the Evaluation of Treated Gliomas. Contrast Media Mol Imaging. 2018;2018:6828396.
  4. Gao Y, Xiao X, Han B, Li G, Ning X, Wang D, et al. Deep Learning Methodology for Differentiating Glioma Recurrence From Radiation Necrosis Using Multimodal Magnetic Resonance Imaging: Algorithm Development and Validation. JMIR Med Inform. 2020;8(11):e19805.
  5. Piper RJ, Senthil KK, Yan JL, Price SJ. Neuroimaging classification of progression patterns in glioblastoma: a systematic review. Journal of neuro-oncology. 2018;139(1):77-88.

Round 2

Reviewer 3 Report

The authors answered all my questions, and I think that the manuscript, in the present form, is now suitab le for publication